# Ganglioside Profiling Uncovers Distinct Patterns in High-Risk Neuroblastoma

**DOI:** 10.3390/ijms26178431

**Published:** 2025-08-29

**Authors:** Claudia Paret, Arthur Wingerter, Larissa Seidmann, Arsenij Ustjanzew, Shobha Sathyamurthy, Jannis Ludwig, Philipp Schwickerath, Chiara Brignole, Fabio Pastorino, Saskia Wagner, Khalifa El Malki, Wilfried Roth, Roger Sandhoff, Jörg Faber

**Affiliations:** 1Department of Pediatric Hematology/Oncology, Center for Pediatric and Adolescent Medicine, University Medical Center of the Johannes Gutenberg-University Mainz, 55131 Mainz, Germany; paretc@uni-mainz.de (C.P.); arthur.wingerter@unimedizin-mainz.de (A.W.);; 2Helmholtz-Institute for Translational Oncology Mainz (HI-TRON), 55131 Mainz, Germany; 3University Cancer Center (UCT), University Medical Center of the Johannes Gutenberg-University Mainz, 55131 Mainz, Germany; 4German Cancer Consortium (DKTK), Site Frankfurt/Mainz, Germany, German Cancer Research Center (DKFZ), 69120 Heidelberg, Germany; 5Institute of Pathology, University Medical Center of the Johannes Gutenberg-University Mainz, 55131 Mainz, Germany; larissa.seidmann@unimedizin-mainz.de (L.S.);; 6Institute of Medical Biostatistics, Epidemiology and Informatics (IMBEI), University Medical Center of the Johannes Gutenberg-University Mainz, 55131 Mainz, Germany; 7Lipid Pathobiochemistry, German Cancer Research Center, 69120 Heidelberg, Germany; pharma.shobha@gmail.com (S.S.); philipp.schwickerath@gmail.com (P.S.); r.sandhoff@dkfz-heidelberg.de (R.S.); 8Laboratory of Experimental Therapies in Oncology, IRCCS Istituto Giannina Gaslini, Via G. Gaslini 5, 16147 Genoa, Italy

**Keywords:** neuroblastoma, GD2, gangliosides, *B4GALNT1*, dinutuximab, naxitamab, ceramide

## Abstract

High-risk (HR) neuroblastoma (NBL) patients often receive standardized treatment despite wide variations in clinical outcomes, underscoring the need for improved stratification tools. A distinguishing feature of NBL is the patient-specific expression of gangliosides (GGs), particularly GD2, which may serve as biomarkers. We analyzed GG profiles in 18 patient-derived tumors and 11 NBL cell lines using thin-layer chromatography and mass spectrometry. Expression of 0-, a-, and b-series GGs was examined and correlated with clinical risk, outcome, and gene expression data. Low-risk (LR) tumors expressed higher levels of complex b-series GGs. In HR tumors, five GG profiles (A–E) were identified. Profile A featured complex b-series GGs; B showed GD2 dominance; C showed synthesis arrest at GM3 or GD3 due to low expression of the GM2/GD2 synthase, encoded by the *B4GALNT1* gene; D included complex a- and b-series GGs; and E was marked by GM2 and GD1a prevalence. *B4GALNT1* expression served as a prognostic marker. Relapsed tumors following anti-GD2 therapy typically exhibited reduced GD2 levels, except for one profile A tumor that displayed a ceramide anchor shorter than those found in LR tumors. Astonishingly, the ceramide anchor composition of GD2 itself appears to separate LR and HR NBL, hinting at a role of ceramide synthases in NBL biology. All cell lines expressed GM2, but exhibited very low levels of complex b-series GGs. Profile C was found only in cell lines of the mesenchymal subtype. These findings support further investigation of GG composition and associated enzyme expression as potential biomarkers for risk stratification and treatment response in NBL.

## 1. Introduction

Neuroblastoma (NBL) is among the most common extracranial solid tumors in children. Due to the significant biological and clinical heterogeneity of this disease, effective risk stratification is crucial for guiding treatment decisions. While over 90% of children with low-risk (LR) neuroblastoma achieve long-term survival with mild or even no treatment, prognosis for those with high-risk disease (HR) remains poor despite intensive multimodal therapy approaches [1,2,3]. The current standard of care for patients with HR NBL involves an intensive treatment regimen that includes chemotherapy, surgical tumor resection, autologous stem cell transplantation, radiotherapy, and immunotherapy with anti-GD2 mAbs [4,5,6].

In recent years, substantial efforts have been directed towards translating genomic and transcriptomic discoveries into novel therapeutic strategies. Despite substantial advancements in the molecular characterization of NBL, few actionable targets have been identified (mainly alterations in the anaplastic lymphoma kinase, encoded by the *ALK* gene), and the survival rates for HR patients have not improved meaningfully. Additionally, immune-based therapies, such as immune checkpoint inhibitors, have shown limited efficacy in NBL, even if some HR patients have a high mutational burden [7]. This is largely due to the unfavorable tumor microenvironment and deficient major histocompatibility complex I (MHC-I) expression [8,9]. Consequently, novel treatment paradigms are needed. Furthermore, most patients with HR NBL are treated uniformly, without additional stratification, despite variability in outcomes. There is an urgent need for new strategies to identify factors that can better stratify these patients, especially to define an “ultra-high-risk” (UHR) subgroup that faces an exceptionally poor prognosis [10].

One peculiarity of the NBL biology is the elevated level of a distinctive class of lipid-linked glycans known as gangliosides (GGs), the most famous representative being GD2 [11]. GGs are amphiphilic glycosphingolipids (GSLs) containing N-acetylneuraminic acid residues (sialic acid). GG synthesis is a multi-step enzymatic process taking place in the endoplasmic reticulum and Golgi apparatus. It begins with the attachment of glucose to ceramide, forming glucosylceramide, after which glycosyltransferases and sialyltransferases sequentially add specific sugar residues and sialic acid, respectively, to generate progressively more complex gangliosides (Figure 1). Gangliosides of the ganglio-series are particularly abundant in the brain and are usually classified in four series (0-, a-, b-, and c-series) according to the number of sialic acid residues on the inner galactose moiety (Figure 1). Their structure and composition are highly variable and this diversity is exploited during embryogenesis, when GGs are produced at specific developmental stages, and to allow cell-to-cell variability in syngeneic cell populations [12]. The GG GD2 is well known to be highly expressed on the surface of most NBL cells, serving as a target for immunologic treatment approaches in HR NBL, including anti-GD2 mAbs (e.g., dinutuximab and naxitamab), chimeric antigen receptor (CAR)-T cells, and vaccines [13,14,15]. GD2 is produced from GD3 through the addition of an N-acetylgalactosamine residue, a reaction catalyzed by the enzyme β-1,4-N-acetylgalactosaminyltransferase, encoded by the *B4GALNT1* gene. *B4GALNT1* expression has been described in the majority of HR NBL cases and has been discussed as a surrogate biomarker of GD2 expression [16]. Its expression has been used in combination with other marker genes for the detection of minimal residual disease in peripheral blood and bone marrow [17]. However, the GG composition of NBL is considerably more complex and highly heterogeneous. Previous studies have linked specific ganglioside alterations to differences in the clinical and biological behavior of NBL [11,18,19]. Deregulated GG expression can indeed support the tumor development by regulating membrane receptors, modulating the expression of MHC-I, as well as inhibiting the activity of immune cells, including T cells and natural killer (NK) cells [20,21]. Moreover, GGs produced by NBL are shed from tumor cells in the tumor microenvironment and blood and have been discussed as a liquid biomarker for patient monitoring [22,23,24].

Here, we analyzed the GG profiles of NBL tissues and cell lines. Our results indicate that GG profiles can distinguish HR from LR NBL and identify HR profiles of diagnostic and therapeutic relevance.

## 2. Results

### 2.1. Patient and Tumor Samples

In total, 18 tumor samples from 16 NBL patients were analyzed by mass spectrometry (Table 1). All eight tumor samples of LR NBL patients, as well as four out of ten HR tumor samples, were obtained at primary diagnosis prior to any systemic treatment. In our analysis, we also included two HR samples, which were collected at primary diagnosis after induction chemotherapy and four samples of either primary refractory or relapsed HR patients after receiving treatment with anti-GD2 mAbs.

### 2.2. The b-Series of GGs Is Associated with LR NBL

We analyzed the expression of GGs of the 0-series (GM1b, GD1alpha, and GD1c), a-series (GM3, GM2, GM1a, and GD1b), and of the b-series (GD3, GD2, GD1b, and GT1b) by mass spectrometry (Figure 2A, Appendix A and Appendix A) in 18 NBL samples (Table 1 and Appendix A). Four tumor samples of HR patients and eight samples of LR patients at primary diagnosis were available. We also included two samples of HR patients, which were obtained after induction chemotherapy, and four samples of HR patients after anti-GD2 therapy with monoclonal antibodies. LR samples significantly expressed more GT1b, GD1b, and GD3, all belonging to the b-series. Some HR samples expressed a high amount of GM2 and GD1a, which belong to the a-series. The expression of GM3 and GD2 was not significantly different between the two groups. The 0-series was not expressed. GT1b exhibited the highest concentration, particularly in the LR group. The concentration of all GGs and of GT1b, but not of GD2, was significantly higher in LR group, regardless of whether the HR samples were retrieved at diagnosis (Figure 2B), following chemotherapy (Figure 2C), or after anti-GD2 therapy (Figure 2D). The difference remained significant even when the outlier LR sample was excluded across samples at diagnosis (all GGs, *p* = 0.0121, GT1b, *p* = 0.0061). For one patient, samples were retrieved after chemotherapy and after anti-GD2 therapy. Of these, the sample retrieved after chemotherapy (#14 in Table 1) had the highest expression in GG, GT1b, and GD2 across the HR samples (Figure 2C, green triangle). However, after treatment with an anti-GD2 mAb, all three values were reduced (red triangle vs. green triangle in Figure 2D, sample #18). For another patient (indicated with a diamond), samples retrieved at diagnosis (#11) and after an anti-GD2 mAb (#17) were analyzed. This sample showed the second highest GD2 expression across the HR samples in the sample at diagnosis, but again, a reduction in GD2 was observed after an anti-GD2 mAb treatment (red diamond vs. green diamond in 2D). The amount of GD2 in HR samples correlated with the level of expression of *B4GALNT1*, which was required for the last step in GD2 synthesis (Figure 1 and Figure 2E, Spearman correlation: r = 0.95). The previously mentioned samples #14 and #11, which exhibited the highest GD2 expression, also demonstrated the highest expression of *B4GALNT1*, while the two HR samples with the lowest GD2 expression had the lowest *B4GALNT1* expression (Figure 2E).

Taken together, these results indicate a clear dominance of the b-series in LR samples and specifically high expression of the complex ganglioside GT1b. GD2 expression is not associated with the risk, but its expression is correlated with the amount of *B4GALNT1*.

### 2.3. GG Profiles of HR Samples

HR patients had very heterogeneous GG profiles. While LR NBL expressed particularly complex GGs, particularly of the b-series, within HR samples, we observed the following profiles (Table 2): (**A**) similar profile as in LR, characterized by the dominance of GT1b (Figure 3A, lane a); (**B**) almost only GD2 is expressed (Figure 3A, lane b); (**C**) only GM3 or GD3 is expressed (Figure 3A, lanes d to f); (**D**) complex GGs of the a- and b-series are both expressed, with a particularly strong expression of GD1a (Figure 3A, lane c); and (**E**) GD1a and GM2 (both a-series) are the prevalent species expressed (Supplemental Appendix A, sample #18). In our small cohort of HR samples, MYCN amplification was not associated with a particular GG profile (Appendix A). In our cohort of high-risk patients, three died of the disease. Only one of them was MYCN-amplified. Two of the patients had tumors characterized by the stopping of synthesis beyond GM3 or GD3 (profile C), with very low GD2 expression, irrespective of the MYCN status (Figure 3B). For the patient with MYCN amplification, samples collected at different time points were available. Interestingly, this profile did not change at different time points of the therapy (Figure 3A, lanes d to f). One HR patient who died of the disease (not MYCN-amplified) had a profile similar to a LR NBL (Figure 3A, lane a). GD2 expression was high in this sample, even though the patient received anti-GD2 mAb, while GD2 expression was rather low in the other samples retrieved after anti-GD2 therapy, suggesting an intrinsic resistance to anti-GD2 mAb therapy (Figure 3C, red point). The composition of the ceramide anchor of GD2 in this sample was different from the one detected in LR NBL. We were indeed able to identify GD2 isoforms with different lengths of the ceramide anchor across the NBL samples. Considering the ceramide anchors of children contain predominantly C18-sphingosine (with a minor contribution of C20-sphingosine), we identified ceramide anchors containing C18 and C20 fatty acid chains (ceramide anchor length of 36–38), typical of normal neurons; shorter anchors containing saturated C14 and C16 (ceramide anchor length of 32–34); and very long ceramide anchors with C22–C26 fatty acids (ceramide anchor length of 40–44). Each sample contained a mixture of GD2 with different lengths of anchor. Some samples expressed a relatively high (about 30%) portion of shorter and very long ceramides (Table 1 group “short”), some samples expressed preferentially normal neuronal ceramides (C36-38) (Table 1 group “normal”), and some samples almost exclusively contained very long ceramides (Table 1 group “long”) (Figure 3D, left panel). When the samples were divided according to the risk, LR samples had a dominance of ceramide anchor length of 36–38 (Figure 3D, right panel). The sample with profile A and high GD2 expression after anti-GD2 mAb treatment was characterized by a high proportion of smaller ceramide anchors (32-34) (Figure 3D, red point in the right panel). Very long ceramide anchors with C22–C26 fatty acids (ceramide anchor length of 40–44) were less abundant in LR samples, and interestingly, all samples with mostly very long ceramide anchors were HR samples only. MYCN amplification was not associated with a particular length of the ceramide anchor (Appendix A), but, again, the limited sample size reduces the statistical power to detect associations.

### 2.4. B4GALNT1 Downregulation as a Predictive Marker of Poor Outcome

The two samples which expressed only GM3 or GD3 (profile C) had the lowest expression of GD2 across the HR tumors and the lowest expression of *B4GALNT1* (Figure 4A, green points), which is required for the synthesis of GM2, GD2, and more complex GGs. Both patients had an extraordinarily aggressive clinical course and died relatively early after diagnosis due to uncontrolled progressive disease. Meta-analysis across three independent RNA-seq datasets (*n* = 124, 95, and 498 with OS) using continuous *B4GALNT1* gene expression as a covariate in Cox proportional hazards models yielded a pooled effect estimate (HR) of 0.67 (95% CI: 0.56 to 0.82, z = −4.04, *p* < 0.0001) under both common effect and random effects models. There was no evidence of between-study heterogeneity (τ^2^ = 0.00 (95% CI 0.00–0.049), I^2^ = 0.0% (95% CI 0.0–89.6), Q(2) = 0.13, *p* = 0.91), noting that precision of heterogeneity estimates was limited with k = 3 (Figure 4B and Appendix A). These results suggest a robust and directionally consistent association between higher *B4GALNT1* expression and improved overall survival across datasets. Unfortunately, we did not have access to datasets containing survival data in which patients were stratified according to anti-GD2 therapy status. As a result, we were unable to directly assess the predictive value of *B4GALNT1* expression for treatment response.

### 2.5. Neuroblastoma Cell Lines Preferentially Express the a-Series of Gangliosides

We also analyzed the expression of GGs in eleven NBL cell lines (Table 3). All cell lines expressed GM2, but the levels of GD1b and GT1b were found to be minimal (Figure 5A). GM2 was more abundant than GD2, except the CHP-134 cell line, which expressed the highest amount of GD2, and the LA-N-5 cell line, which demonstrated a comparable level of GD2 and GM2. Accordantly, the a-series GGs dominated over b-series GGs, with the exception of the CHP-134 cells, which demonstrated a high expression of GD2 and GD3; the LA-N-5 cells, which exhibited the lowest level of GM2 and the highest level of GT1b; and the ACN cells, which expressed a high amount of GD3 (Figure 5B). Most cell lines preferentially expressed simple rather than complex GGs, with a clear exception of the murine cell line NXS2 that expressed a high amount of GD1a. Compared to the profiles found in patient-derived NBL samples, LA-N-5 could be partially aligned to profile A (presence of GT1b, although the expression was not higher than GM2). Profile B (dominance of GD2) was found in the cell line CHP-134. Profile C (prevalence of GM3 and/or GD3) was found prevalently in GI-ME-N and ACN, followed by SK-N-AS (Figure 5C). Profile D (complex GGs of the a- and b-series were both expressed) was not found in any cell line. Profile E (prevalence of GD1a and GM2) was found in NBL-S, SK-N-BE(2)-C, SH-SY5Y, and NXS2, while LS had the highest amount of GM2, but a low concentration of GD1a and IMR-32 expressed almost exclusively GM2. The cell line SH-SY5Y NBL exhibits two distinct lineage states characterized by their cellular phenotypes and differentiation status [27]. The first is a differentiated adrenergic state, which resembles mature sympathetic neurons. The second is a mesenchymal state, often described as ‘neural crest cell-like’, which is less differentiated and displays features associated with neural crest progenitors, including increased migratory capacity and plasticity. Interestingly, profile C was observed only in the mesenchymal subtype (mes) and in 3 out of 4 mes cell lines (Table 3). Based on transcriptome data of publicly available NBL cell lines, *B4GALNT1* downregulation, which could explain the dominance of GM3 and GD3, has been described in GI-ME-N and SK-N-AS (Appendix A). According to these data, another cell line with low expression of *B4GALNT1* is cogn519. Taken together, these results indicate that NBL cell lines have a profile reminiscent of HR NBL, with a prevalence for simple GGs of the a-series, and the low expression of complex GGs of the b-series. However, the high heterogeneity of the profiles makes it difficult to correlate each one with profiles found in the clinical sample.

## 3. Discussion

NBL can be viewed as a failure in the proper differentiation of neural crest cells. These cells typically differentiate into neurons of the sympathetic ganglia or sympathetic neuron-like chromaffin cells of the adrenal medulla or Schwann cells in response to specific signals. During this differentiation process, NBL cells emerge, which may resemble immature sympathetic neurons (N-type cells) or Schwann cells (S-type cells) [42]. In neurons, the total GG content increases several-fold from the embryonic stage to the postnatal life, with a parallel shift from simpler (i.e., GD3 and GM3) to more complex GG species (i.e., GM1, GD1a, GD1b, and GT1b) [43]. Therefore, it is reasonable to infer that the GG composition in NBL reflects the developmental stage of the neural crest cells from which the tumor cells originate. Given the crucial role of GGs in regulating various processes, including differentiation and immunomodulation, it is expected that GG composition can significantly impact the biology and treatment of NBL [44]. Our findings suggest that the composition of GGs may influence both patient risk stratification and therapeutic response.

In terms of stratification, we observed a clear difference between the GG profiles in LR and HR tumor samples. LR NBL is characterized by a high abundance of complex GGs, with a predominance of the b-series over the a-series. This pattern is reminiscent of the GG profile observed during mid-embryogenesis, where GT1b is initially expressed at higher levels than GD1a. In contrast, in the mature brain, both GT1b and GD1a are expressed at comparable levels [43]. Indeed, during the development, GD3-synthase (encoded by *ST8SIA1*) slightly decreases, and as a result, more GD1a is synthesized in late embryonic, postnatal, and adult brains [43]. Correlation between LR and high amounts of complex b-series GGs was suggested previously by using thin-layer chromatography [45]; with our method we quantified ten GGs with high specificity. We observed five GG profiles in HR samples. Of particular interest were the samples expressing only the simple GGs GM3, and GD3 (profile C), which are the main GGs of neural stem cells [43]. This profile was found in patients with an exceptionally severe clinical course who died of uncontrolled progressive disease early after primary diagnosis in our small cohort. Therefore, the biosynthetic stop beyond GM3 and GD3 with consequent minimal GD2 expression may characterize a highly aggressive, immature NBL subtype, constituting an ultra-high-risk cohort, resistant to GD2-targeted immunotherapeutic approaches. This finding should be necessarily validated in a larger cohort. Further studies are also needed to determine whether GM3/GD3 enrichment is indicative of cancer stem cell-like characteristics in NBL. These samples were characterized by a very low expression of *B4GALNT1*, which is required for the synthesis of GD2 and GM2 and therefore of more complex GGs. Loss of *B4GALNT1* expression correlated with worse survival in several, but not all, available cohorts with transcriptome or microarray data. Loss of *B4GALNT1* expression could therefore be used as a biomarker to identify patients with particularly aggressive tumors, but should be validated in characterized cohorts. We also observed a clear difference in the ceramide length between HR and LR tumors. The de novo synthesis of ceramides is catalyzed by six (dihydro)ceramide synthases (CerS1–6), each with specificity for different acyl-CoA substrates. Stearic acid (C18:0), the main fatty acid of GGs in LR NBL, is also the main fatty acid of GGs of adult differentiated neurons, due to the dominant expression of ceramide synthase 1 (CerS1) [46]. Shorter ceramide reflects the expression of CerS5 (or 6), while very long ceramide reflects the expression of CerS2. Ceramide synthases are differentially expressed in the developing brain, with CerS1 increasing and CerS6 decreasing during brain development [47]. CerS2 is strongly expressed in oligodendrocytes and Schwann cells and catalyzes the synthesis of very long acyl chain-containing ceramides, which generate sphingomyelin and cerebrosides [48]. Interestingly, mesenchymal NBL tumor cells which resemble Schwann cell precursors of the developing neural crest are enriched in relapsed tumors [49]. A correlation between CerS1/5/6 expression and ceramide length in NBL, as well as the association of CerS expression with prognosis, should be analyzed in larger cohorts. In terms of clinical utility and therapy management, some profiles could help to predict sensitivity to retinoid therapy and to anti-GD2 targeted therapies. Retinoid-based tumor differentiation therapy has a well-established history in clinical development and is recognized as a standard treatment for HR NBL in some protocols [50]. However, there are still many HR NBL patients who have tumors that do not respond to retinoid-based therapy. Neuronal differentiation in vitro can be induced by exposure of NBL cells to exogenous complex gangliosides, particularly GT1b and GQ1b [51]. Interestingly, the complex GGs of the a- and b-series, along with a more differentiated phenotype, can be induced by all-trans retinoic acid (ATRA) and 13-cis retinoic acid (RA) in several, but not all, NBL cell lines [52]. ATRA can cause a marked increase in GG content, with relative enhancement in GD1a and GT1b synthesis [53]. This induction occurs through the activation of GD1b/GM1a synthase activity, but not GD3 or GD2/GM2 synthase [52]. Retinoic acid can activate the transcription of these glycosyltransferases or stabilize their expression [54]. Complex GGs such as GM1a promote neurotrophin tyrosine kinase receptor A (TrkA)-mediated NBL cell differentiation [55]. Therefore, in tumors with the profile C, where the synthesis of GGs stops at GM3 and GD3, ATRA or 13-cis retinoic RA are unlikely to have any effect. GM2 and GD2 are indeed essential to allow the production of more complex gangliosides, which in turn are required to promote TrkA-mediated NBL cell differentiation [55]. In contrast, in samples with the prevalence of GD2 over all other GGs (profile B), sensitivity to ATRA or 13-cis retinoic RA can be expected. GD2 is an intermediate product during the synthesis of more complex GGs [43], and its accumulation suggests a low activity of the GD1b/GM1a synthase, which could be stimulated by ATRA or 13-cis retinoic RA. However, inhibition of GG synthesis did not block neurite formation induced by ATRA in the LAN-5 cell line [56]. Further mechanistic studies will be needed to assess the relevance of GG profiles in resistance to retinoid therapy.

Concerning anti-GD2 therapies, resistance can be expected in samples with profile C and profile E, which both expressed a very low amount of GD2, either because of the low expression of *B4GALNT1* (profile C) or because the synthesis of GGs was shifted to the a-series (profile E). The shift to the a-series could be due to a downregulation of the gene coding for the GD3-synthase (*ST8SIA1*). Downregulation of *ST8SIA1* has been previously suggested as being associated with the transition to a mesenchymal state resulting in resistance to anti-GD2 mAbs [49]. The a-series is very strong at birth, but diminishes later [57]. Therefore, the high concentration observed at high risk may reflect a particularly proliferative stage of neuronal development. Expression of GM2 has been associated with resistance to anti-GD2 therapy in previous studies, leading to the development of anti-GM2 CAR-T cells [58]. While a low amount of GD2 has been discussed as a reason for resistance to anti-GD2 mAbs [59], only 10–12% of NBL patients show complete or partial loss of GD2 expression in bone marrow-derived NBL cells [60], a percentage far lower than the overall rate of non-response to antibody therapy. Therefore, constitutively low or absent GD2 expression can account for only a portion of anti-GD2 directed therapy resistance. Indeed, a recent study using anti-GD2 CAR-T cells found no loss of GD2 expression on relapsed tumor cells after treatment failure [14]. In our cohort, one sample from a patient who died of the disease exhibited a profile similar to that of an LR NBL, with a high amount of the complex GGs of the b-series GT1b (profile A). This sample showed a high level of GD2 expression, despite being retrieved after growth under anti-GD2 therapy. Interestingly, GD2 in this sample had a significantly higher proportion of the shorter ceramide anchors than found in GD2 of LR samples. GGs that contain shorter-chain fatty acids are known to be preferentially shed by tumor cells, including GD2 and GT1b [23,61]. GD2 ganglioside with a shorter fatty acyl chain is up to 10-fold more active than those with a longer fatty acyl chain in immunosuppression [62]. Several mechanisms of immunosuppression by GGs have been described. GGs shed in serum, including GM3, GD3, GT1b, and GD2, inhibiting the cytotoxicity of NK cells [63]. GGs, particularly GD3 and GT1b but not GD1a, strongly bind siglec-7 [64]. Siglec-7 is mainly expressed on NK cells and mediates negative regulatory signals [65,66]. NBL cells, through GGs, can engage Siglec-7 to inhibit immune responses [67]. GT1b can also bind Siglec-5, which is an inhibitory immune checkpoint molecule for human T cells [68,69]. Of note, as discussed in the literature, circulating GD2 binding to dinutiximab in plasma could block the antibody’s therapeutic effect by saturating the antibody binding sites or potentially inducing systemic complement activation, therefore increasing toxicity [70]. The length of the ceramide can moreover influence the antigenicity of GGs. Particularly, elongation of the fatty acid to 24 carbon atoms, as detected in some HR samples, can reduce the binding affinity of monoclonal antibodies to half [71]. GD2 isoforms are not routinely analyzed in clinical samples. In future studies, more samples should be examined to determine whether these isoforms are associated with resistance to anti-GD2-directed therapies. Moreover, mechanistic testing in the future would be necessary to verify how specific GGs, including GD2 with different ceramide lengths, drive aggressiveness or therapy resistance in NBL.

All cell lines analyzed in this work showed a low concentration of complex GGs of the b-series (GD1b and GT1b) and a prevalence of GGs of the a-series, particularly GM2. This GG composition has been described in several human NBL cell lines [72,73]. Importantly, three of the five profiles (B, C, and E) identified in HR tumor samples are also present in the cell lines, reinforcing their relevance as models for preclinical research. None of the cell lines analyzed in this work can recapitulate profile A (prevalence of complex GGs of the b-series) and D (co-expression of complex GGs of the a- and of the b-series). To the best of our knowledge, the expression of complex b-series GGs, without co-expression of complex a-series GGs, has not yet been reported in NBL cell lines in the literature. Co-expression of complex GGs of the a- and b-series (profile D) has been described in the cell lines KP-N-NS and SMS-KCN and in the SH-SY5Y cell line upon retinoic acid-induced neuronal differentiation [72,73,74]. About forty NBL cell lines have been described, but only a subset have been characterized for their GG profiles. Further GG profiling is needed to identify cell lines that match all GG profiles observed in patient samples. Such cell lines would enhance the available in vitro models, particularly for studying aggressive NBL subtypes [75]. Based on our results, the GIMEN cell line could, for example, represent a good model for the analysis of NBL clinical samples with high expression of GM3 (profile C). Given that patients with this profile had a particularly dismal prognosis, this model could be used to analyze the induction of GD2 and of more complex GGs. Interestingly, all NBL cell lines with profile C were of the mesenchymal subtype, which is characterized by a more aggressive phenotype. GGs interact with cell surface receptors, modulating signaling pathways involved in cellular processes such as growth, differentiation, and apoptosis. As different GG structures may have opposite regulatory effects on these receptors, it would be important to include GG profiles in the characterization of preclinical models of NBL [76]. Moreover, the composition of GGs is affected by the growth conditions. Particularly, the a-series is abundant when the cells are grown in cell culture, while the same cells grown in mice express more of the b-series [77].

A key limitation of this study is the relatively small number of NBL samples analyzed, as well as their heterogeneity. The limited sample size reduces the statistical power to detect associations, while the variability in clinical and molecular characteristics among the samples introduces potential confounding factors that may obscure or bias the observed results. Consequently, the findings should be interpreted with caution, and future studies should aim to include larger, more homogeneous cohorts to validate and extend these results. Although we were able to identify a clear distinction between HR and LR samples, several distinct profiles were observed within the HR group. The prevalence of these profiles within the broader NBL population still needs to be determined. Because GG analysis is not routinely performed, the expression of genes, such as *B4GALNT1*, or of multiple genes involved in the GG network, could help to identify and characterize such subtypes [19].

## 4. Materials and Methods

### 4.1. Patients and Tissues

Surplus fresh frozen tissues from surgery were used for the analysis of gangliosides. Tumor areas were isolated by an experienced pathologist. In accordance with the ethics committee of Rhineland–Palatinate, written informed consent of all patients or their custodians was obtained for “scientific use of surplus tissue not needed for histopathological diagnosis”. Diagnosis and treatment-related data were extracted from archived medical records. Risk stratification of NBL patients was performed according to national guidelines [4]. This study was performed in agreement with the Declaration of Helsinki on the use of human material for research. Ethical approval was obtained by the local ethics committee (No. 2020-15404, 16 November 2020 and No. 2021-15871, 19 May 2021). Transcriptome and clinical data were available via the R2: Genomics Analysis and Visualization Platform (http://r2.amc.nl), accessed in June 2025.

### 4.2. Cell Cultures

Human neuroblastoma cell lines were kept at 37 °C and 5% CO_2_ in 10 cm2 cell culture dishes (SARSTEDT AG & Co.KG, Nümbrecht, Germany). The LS and CHP-134 (provided by DMSZ, Braunschweig, Germany) and IMR-32, SK-N-AS, SK-N-BE(2)C, GI-ME-N, LAN-5, and ACN cell lines (provided by Tumor Cell Lines Repository (TCLR) at Laboratory of Experimental Therapies in Oncology, IRCCS Istituto G. Gaslini, Genoa, Italy) were cultivated in RPMI-1640 medium (Sigma Life Science, St. Louis, MO, USA). NBL-S (provided by DMSZ, Braunschweig, Germany) cells were cultivated in IMDM (Sigma Life Science, St. Louis, MO, USA). The SH-SY5Y (provided by DMSZ) and NXS2 cell lines (provided by TCLR) were cultivated in Advanced DMEM high-glucose medium (Sigma Life Science, St. Louis, MO, USA). All media were supplemented with 1% penicillin/streptomycin (Dickson and Company, Franklin Lakes, NJ, USA) 1% L-glutamine (Sigma Life Science, St. Louis, MO, USA), and 10% fetal bovine serum (Thermo Fisher Scientific GMbH, Schwerte, Germany). Cells were split twice a week by a confluence of about 85%. Cell pellets of 5 × 10^6^ cells were frozen at −80 °C without medium for mass spectrometry. Cell lines were periodically tested for mycoplasma contamination by polymerase chain reaction (PCR) assay, characterized by cell proliferation and morphology evaluation, and authenticated by multiplex short-tandem repeat profiling (PowerPlex Fusion-24 loci, Promega, Milano, Italy).

### 4.3. Lipid Extraction

Tissues and cell lines were extracted with solvent mixtures of chloroform–methanol–water, desalted with C18-columns, and split into neutral and acidic lipids with DEAE ion exchange columns as described in detail in [18]. Shortly, freeze-dried tissues were powdered, and up to circa 20 mg dry weight was extracted three times with chloroform–methanol–water. Supernatants were collected, and the residual pellet was used for BCA-based protein quantification. Lipid extracts were desalted on C18 columns, separated into neutral and acidic lipids via DEAE ion exchange, and desalted again on C18. Fractions were dried under nitrogen at 37 °C and re-dissolved in chloroform–methanol–water (10:10:1) to 4 mg protein/mL.

### 4.4. Thin-Layer Chromatography and Immune Overlay

Lipids (300 µg protein) were applied to HPTLC plates (Linomat IV, Camag, Muttenz, Switzerland) and developed with a pre-run solvent (chloroform–acetone, 1:1), dried, then run in chloroform–methanol–0.2% aqueous CaCl_2_ (45:45:10) as described in detail in [18]. Detection of ganglioside GD2 was performed using the immuno-overlay technique using an anti-GD2 antibody (BD Pharmingen, San Diego, CA, USA #554272, clone 14.G2a), followed by alkaline phosphatase-conjugated secondary antibody and BCIP/NBT substrate (SigmaFAST, Sigma-Aldrich, Taufkirchen, Germany). In order to visualize all GGs, TLC plates were destained in acetone afterwards. Subsequent to drying, plates were sprayed with orcinol reagent using a Derivatizer from Camag (Muttenz, Switzerland), and developed at 120 °C as described in detail in [18].

### 4.5. LC-MS^2^ Analysis of Gangliosides

An aliquot of the acidic lipid extract corresponding to 600 µg sample protein was mixed with isotopically labeled GG in 100 µL of chloroform–methanol–water (10:10:1) and diluted with 100 µL of solvent A (75% acetonitrile, 19.25% isopropanol, 4% water, 1% methanol, 0.75% DMSO, 0.2% formic acid + 10 mM ammonium acetate). The mixture contained D5-GM3(D5-d36:1) (Avanti Polar Lipids, Alabaster, AB, USA) to quantify GM3, D3-GM2(D3-d36:1 (and D3-d38:1)) (Cayman Chemical Company, Ann Arbor, MI, USA) to quantify GM2, D5-GM1a(D5-d36:1) (Avanti Polar Lipids, Alabaster, AB, USA) to quantify GM1a and GM1b, and D3-GD3(D3-d36:1) (Cayman Chemical Company, Ann Arbor, MI, USA) to quantify GD3, GD2, GD1a, GD1b, GD1c, GD1α, and GT1b; the latter due to the lack of corresponding isotopically labeled standards at that time. For GD2-, GD1-, and GT1-species correction factors for mass spectrometric response differences were taken into account, which were obtained with corresponding unlabeled standards. A volume of 10 µL spiked sample was injected onto an Aqcuity I class UPLC (Waters, Milford, MA, USA) equipped with an ACQUITY UPLC Glycan BEH Amide column (1.7 µm beads with 130 Å, 150 mm × 2.1 mm, Waters, Milford, MA, USA) and run at 40 °C. An HILIC gradient was applied, starting with 100% solvent A and increasing finally to 100% solvent B (10% acetonitrile, 15% isopropanol, 20% water, 54.25% methanol, 0.75% DMSO, 0.9% formic acid + 10 mM ammonium acetate) following re-equilibration: 0/100, 1/100, 2/95, 3.5/90, 6.5/60, 15/0, 19/0, 22/50, 25/100, and 30/100 [min/%A]. GGs were monitored with a triple-quadrupole-like tandem mass spectrometer (Xevo TQ-S, Waters, Milford, MA, USA) in the negative electrospray multiple reaction monitoring mode. The transition of single deprotonated molecular ions of GM3, GM2, GM1a, GM1b, and GD3 and of double deprotonated molecular ions of GD2 and GD1alpha, GD1a, GD1b, and GT1b to the sialic acid fragments of N-acetyl neuraminic acid (Neu5Ac, fragment *m/z* −290.1) for all gangliosides and of N-glycolyl neuraminic acid (Neu5Gc, fragment *m/z* −306.1) for GM3 and GM2 only were monitored. The MRM transitions can be found in Appendix A. The separation capacity of this LC-MS^2^ method is documented in Appendix A. Parameters for mass spectrometric quantifier and qualifier transitions are summarized in Appendix A. GGs with the ceramide anchors 32:1;O2, 34:1;O2, 36:1;O2, 38:1;O2, 40:1;O2, 41:1;O2, 42:2;O2, 42:1;O2, 43:1;O2, 44:2;O2, and 44:1;O2 were taken into account. GM3, GM2, GM1a, and GD3 were quantified in relation to their respective deuterated internal standards. GM1b was quantified in relation to D5-GM1a. GD2, GD1alpha, GD1a, GD1b, and GT1b were quantified in relation to D3-GD3, taking a response factor into account, which was calculated with known concentrations of corresponding standard GGs.

### 4.6. Gene Expression Analysis of Tumor Samples

RNA was extracted from fresh-frozen tumor specimens using the RNeasy Lipid Tissue Kit (QIAGEN, Hilden, Germany) according to the manufacturer’s instructions. The RNA quality was assessed with a Bioanalyzer Device (Agilent Technologies, Santa Clara, CA, USA), and only samples exhibiting sufficient RNA integrity number (RIN) values were selected for further experiments. Reverse transcription was carried out using the PrimeScript™ RT Reagent Kit with gDNA Eraser (TaKaRa BIO INC, Kusatsu, Japan). Quantitative RT-PCR was performed with the PerfeCTa^®^ SYBR^®^ Green Fast Mix^®^ (Quantabio, Beverly, CA, USA) on a LightCycler 480 system (Roche, Basel, Switzerland). Relative quantification was normalized to the housekeeping gene *HPRT1* and calculated using the 2^−ΔΔCt^ method. The calibrator was defined as the maximal PCR cycle number minus the average Ct value of *HPRT1*. The primers used for qRT-PCR were as follows: *B4GALNT1* forward: 5′CCTTCAGGCAGCTTCTGGT3′, reverse: 5′TGCTGTGTTGGTCTGGTAGC3′; and *HPRT1* forward: 5′-TGACACTGGCAAAACAATGCA, reverse: 5′-GGTCCTTTTCACCAGCAAGCT. Data visualization was performed using GraphPad Prism (version 7.02, Boston, MA, USA).

### 4.7. GG Transcriptome Profiles of NBL Cell Lines

The R2: Genomics Analysis and Visualization Platform (https://r2.amc.nl) was employed for the profile gene expression of selected GGs for NBL cell lines from (1) dataset GSE89413 (*n* = 41) [75] and (2) dataset “Cell line Neuroblastoma—Molenaar” (*n* = 20) [78].

### 4.8. Statistical Analysis

HR and LR samples were compared using the Mann–Whitney test. The relationship between two variables was analyzed by Spearman correlation. Survival analysis based on *B4GALNT1* gene expression was conducted using Cox proportional hazards regression via the R2: Genomics Analysis and Visualization Platform (https://r2.amc.nl) across multiple neuroblastoma (NBL) datasets (see Appendix A), accessed on 5 June 2025. Overall survival data were used as the endpoint. The derived *p*-values from the R2 Cox hazard regression, where *B4GALNT1* expression was used as a continuous coefficient, were further used for a pooled meta-analysis to estimate the common and random effects of three RNA-seq studies (EGAD00001006625 (Westermann) [79], and GSE181582 (Bell) [80] and GSE62564 (SEQC) [81] via the R package meta (version 8.1-0) [82]. We restricted EGAD00001006625 to diagnostic (“initial”) tumors for the primary meta-analysis to avoid bias from post-treatment/relapse sampling. Treatment status at sampling was not available for the other two studies.

## 5. Conclusions

In conclusion, our data suggest that the dysregulation of GGs in NBL offers a promising avenue for both diagnosis and therapy. Specifically, the GM3/GD3 profile, associated with very immature and aggressive tumor cells, warrants further attention and validation in a larger cohort. Targeted differentiation of these immature precursor cells could open new therapeutic possibilities. Additionally, our study highlights potential mechanisms of resistance to anti-GD2 therapies, such as shifts to other GG series or the presence of shorter GD2 species. Incorporating GG profiling could enhance risk stratification, support the development of more targeted treatment strategies, and help to identify relevant preclinical models.

## Figures and Tables

**Figure 1 ijms-26-08431-f001:**
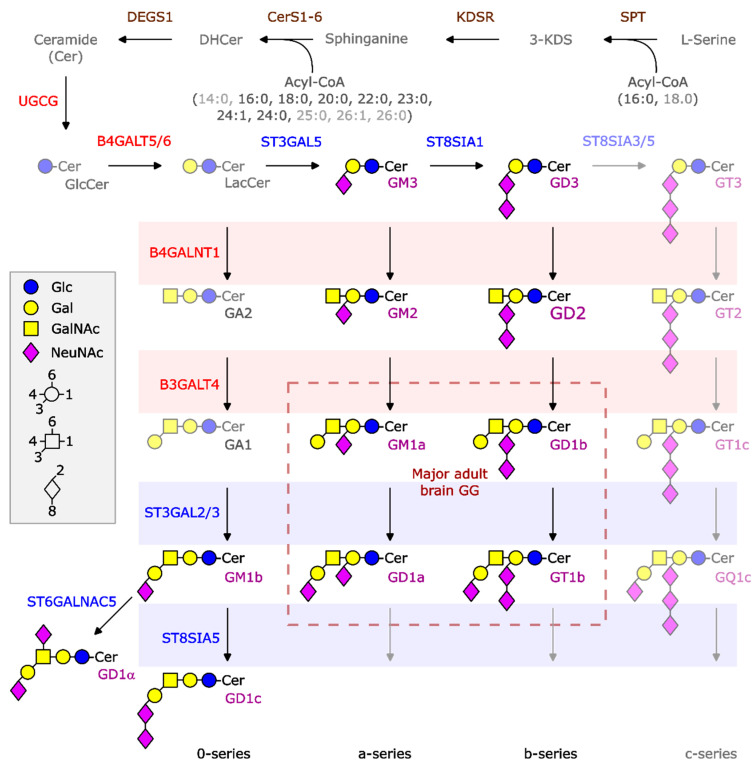
Symbolic structures and biosynthesis of ganglio-series gangliosides (GGs). GGs are synthesized by the stepwise addition of monosaccharides to ceramide (Cer) via the action of different glycosyltransferases (in red) and different sialyltransferases (in blue). Cer is produced in the endoplasmic reticulum by the sequential action of serine palmitolyltransferase complex (SPT), ketodihydrosphingosine reductase (KDSR), one of six ceramide synthases (CerS1-6), and dihydroceramide desaturase 1 (delta 4-desaturase, sphingolipid 1, DEGS1). Transported to the Golgi apparatus, UDP-glucose ceramide glucosyltransferase (UGCG) will convert Cer to GlcCer at the cytosolic leaflet. On the luminal side of the Golgi apparatus, glycosyltransferases and sialyltransferases then can convert GlcCer in a stepwise manner to lactosylceramide (LacCer) and more complex glycosphingolipids (GSLs) and gangliosides. Lac-Cer is substrate for six transferases enabling the production not only of ganglio-series GSLs (as presented), but also of (neo)lacto-, globo-, isoglobo-series and complex sulfatides (not shown here). Gangliosides of the ganglio-series are divided into four series. The “0-series” includes GM1b, GD1alpha, and GD1c. The “a-series” includes GM3, GM2, GM1a, and GD1a. GD3, GD2, GD1b, and GT1b belong to the “b-series” and GT3, GT2, GT1c, and GQ1c belong to the “c-series”. The arrows indicates that the synthesis can continue towards more complex structures. In normal brains, basically, a- and b-series GGs are expressed. The length of the amide-bound acyl chain (see indication in brackets) depends on the specificity of the expressed ceramide synthases (CerS1-6) and is dominated by CerS1 and the presence of a stearic acid residue in normal brain neurons [25]. Glc: glucose, Gal: galactose, GalNAc: N-acetyl galactosamine, NeuNAc: N-acetyl neuraminic acid, 3-KDS: 3-ketodihydrosphingosine, DHCer: dihydro-ceramide, B(3/4)GALT4/5/6: beta-1-(3/4)-galactosyltransferase 4/5/6, B4GALNT1: beta-1-4-N-acetyl galactosaminyltransferase 1, ST3GAL2/3/5: ST3 beta-galactoside alpha-2,3-sialyltransferase 2/3/5, ST6GALNAC5: ST6 N-acetyl-galactosaminide alpha-2,6-sialyltransferase 5, ST8SIA1/3/5: ST8 alpha-N-acetyl-neuraminide alpha-2,8-sialyltransferase 1/3/5.

**Figure 2 ijms-26-08431-f002:**
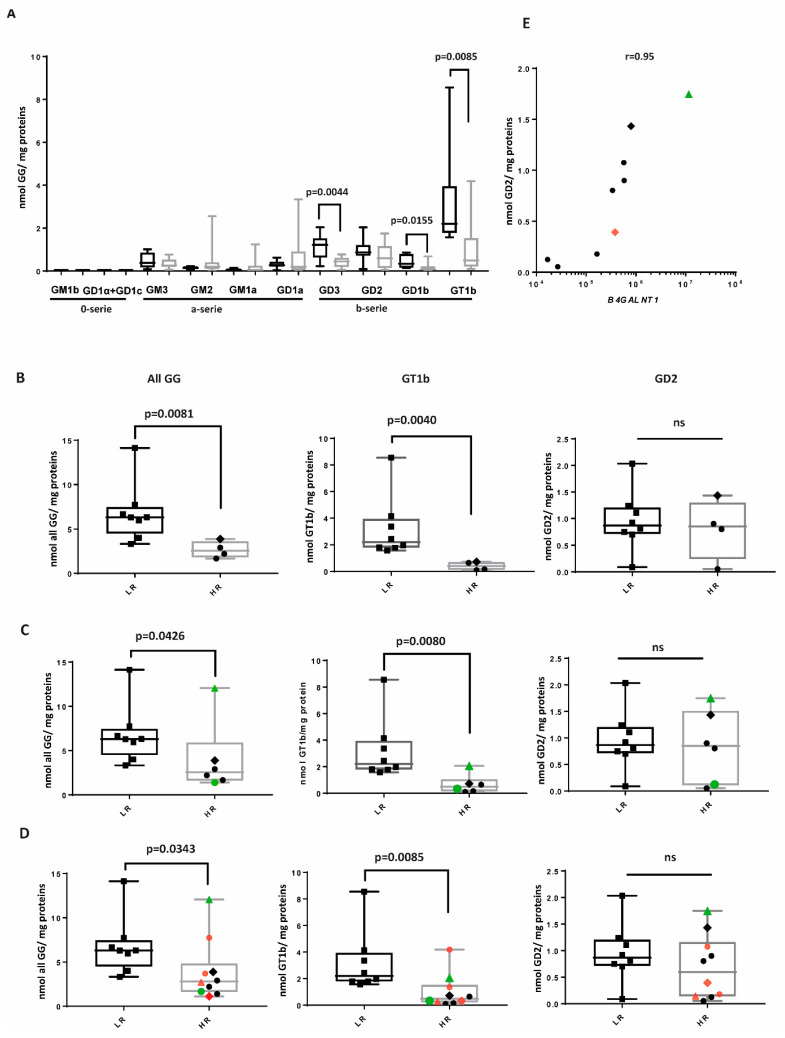
The b-series of GGs is associated with low-risk NBL. The GG composition of eight low-risk (LR) (black box plots) and ten high-risk (HR) samples (gray box plots) was analyzed by mass spectrometry in all samples (**A**), only in samples at diagnosis (**B**), or after inclusion of samples (in green) retrieved after chemotherapy (**C**) and samples (in red) retrieved after anti-GD2 therapy (**D**). The difference in the expression levels of all GGs, GT1b, and GD2 was assessed using a Mann–Whitney test. The relationship between the amount of GD2 in nine HR samples and the expression of *B4GALNT1* is shown in (**E**) and was analyzed using Spearman correlation (monotonic relationship). The samples indicated with a triangle or with a diamond, respectively, in (**C**), (**D**), and (**E**) belong to the same HR patients.

**Figure 3 ijms-26-08431-f003:**
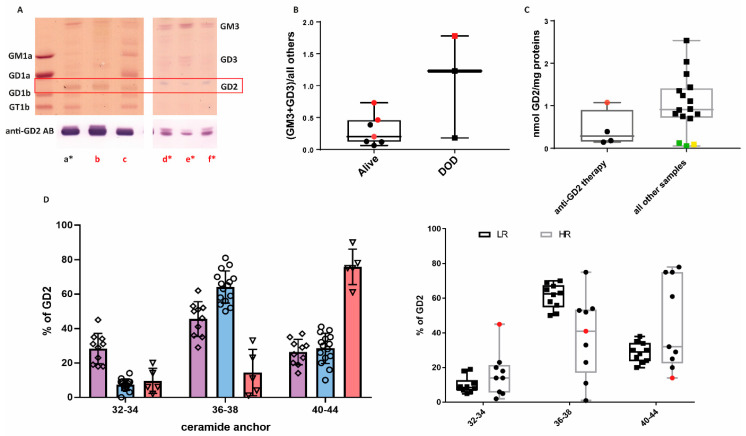
GG profiles of high-risk (HR) samples. (**A**) GGs were extracted and analyzed by thin-layer chromatography in six HR NBL samples (a to f). Samples d to f belong to the same patients and were retrieved at diagnosis (d) or after 5 cycles (e) or 6 cycles (f) of chemotherapy. GGs were stained with orcinol (upper panel) or with an antibody (AB) against GD2 (lower panel). A mixture of GGs isolated from beef brain (GM1a, GD1a, GD1b, and GT1b) was used as reference. MYCN-amplified samples are in red. Samples of patients who died of the disease are marked with an asterisk. The GD2 band is marked with a red box. (**B**) The ratio between the sum of GM3 and GD3 and the sum of all other GGs (GM3 + GD3/all others, Y axis) was calculated in HR patients who survived (alive) or died of the disease (DOD). MYCN-amplified samples are in red. (**C**) GD2 expression in samples retrieved from four patients after anti-GD2 mAb treatment. The sample in red has GG profile A; the samples in green have profile C. The sample in yellow is a low-risk (LR) sample. (**D**) Length of the ceramide anchor of GD2 determined by mass spectrometry. The length is indicated as the sum of the sphingoid base, considered to be mainly C18, and the length of the fatty acid chain (X-axis). The Y-axis indicates the % of GD2 with the respective ceramide anchor. Each sample contains a mixture of GD2 species with varying ceramide anchor lengths. In the left panel, samples in violet (diamond) expressed a relatively high portion of shorter and very long ceramides; samples in blues (circles) contained predominantly normal neuronal ceramides; and samples in red (triangle) expressed almost exclusively longer ceramides. In the right panel, samples were divided into low-risk (LR) and high-risk (HR). The sample in red was retrieved after anti-GD2 therapy and had high GD2 expression. Note that all four samples with high C40–44 ceramide anchor content belonged to the HR group.

**Figure 4 ijms-26-08431-f004:**
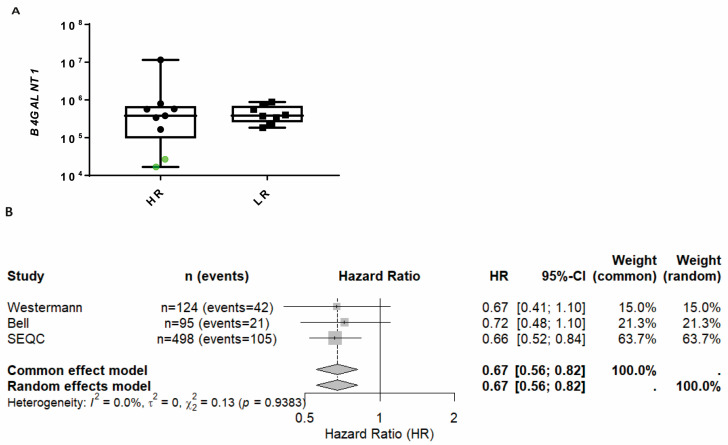
*B4GALNT1* downregulation as a predictive marker of poor outcome. (**A**) The expression of genes required in the GG synthesis were analyzed by qRT-PCR in high-risk (HR) and low-risk (LR) samples. The samples in green belong to two patients with profile C. (**B**) The meta-analysis was performed with publicly available transcriptome and survival data. The number of patients with available survival data was 124, 95, and 498 in the Westermann, Bell, and SEQC studies, respectively. The Westermann dataset was restricted to samples collected at diagnosis only (disease class: initial); Bell and SEQC did not report treatment status at sampling. The random effects model used the DerSimonian–Laird estimator for τ^2^; squares show study’s hazard ratio (HR) from Cox models with *B4GALNT1* as a continuous covariate; diamond shows pooled estimate.

**Figure 5 ijms-26-08431-f005:**
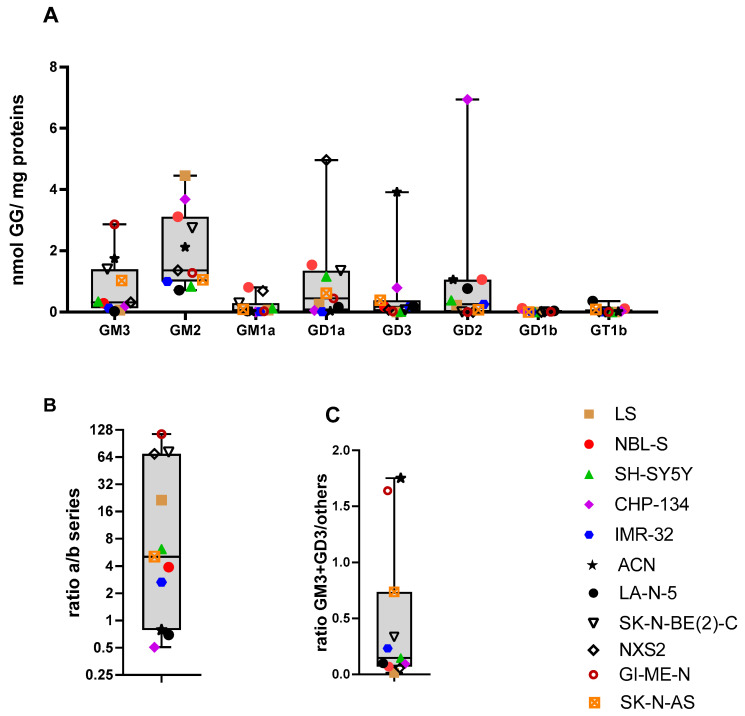
Neuroblastoma cell lines preferentially express the a-series of gangliosides. GGs of the a- and of the b-series were quantified by mass spectrometry in eleven cell lines. Each cell line is indicated by a symbol; the distribution of the respective values is shown as box and whisker plots. (**A**) Concentration of the indicated GGs (X-axis) across the cell lines. The amount of each GG is indicated as nmol GG per mg of total protein (Y-axis). (**B**) The ratio between the sum of GGs of the a-series (GM3, GM2, GM1a, GD1a) and the sum of GGs of the b-series (GD3, GD2, GD1b, GT1b) was calculated for each cell line (Y-axis: ratio a/b-series). (**C**) The ratio between the sum of GM3 and GD3 and the sum of all other GGs was calculated for each cell line (Y-axis: ratio (GM3+GD3)/others).

**Table 1 ijms-26-08431-t001:** Neuroblastoma patients and sample characteristics.

Patient ID	Risk	Last Therapy Before Sample Collection	Age at Diagnosis	Sex	Histology	Sample Localization	Disease Status at Sample Collection	MYCN	GG Profile	Ceramide
#1	LR (4S)	none	8 m	m	NBL, pd	Metastatic (skin, lumbar)	Primary disease	NA	A	normal
#2	LR (4S)	none	2 m	m	NBL, pd	Primary tumor (adrenal gland)	Primary disease	NA	A	normal
#3	LR	none	5 m	f	NBL, pd	Primary tumor (adrenal gland)	Primary disease	NA	A	normal
#4	LR	none	1 m	f	NBL, pd	Primary tumor (adrenal gland)	Primary disease	NA	A	normal
#5	LR	none	4 m	f	NBL, pd	Primary tumor (adrenal gland)	Primary disease	NA	A	normal
#6	LR	none	4 y	f	NBL, dif/mature GN	Primary tumor (adrenal gland)	Primary disease	NA	A	normal
#7	LR	none	5 m	m	NBL, pd	Primary tumor (adrenal gland)	Primary disease	NA	A	normal
#8	LR	none	9 m	m	NBL, pd	Primary tumor (retroperitoneal)	Primary disease	NA	A	normal
#9	HR	none	1 y	m	NBL, pd	Metastatic (lymph node, supraclavicular)	Primary disease	A	B	normal
#10	HR	none	2 y	m	NBL, pd	Primary tumor (adrenal gland)	Primary disease	A	C	long
#11	HR	none	8 y	m	NBL, pd	Metastatic (lymph node, cervical)	Primary disease	NA	B	long
#12	HR	none	1 y	m	NBL, pd	Primary tumor (retroperitoneal)	Primary disease	A	B	short
#13	HR	chemotherapy	4 y	f	NBL, pd	Metastatic (orbital)	Primary refractory, PD	NA	C	long
#14	HR	chemotherapy	8 y	m	NBL, dif	Primary tumor (retroperitoneal)	Primary disease	NA	D	normal
#15	HR	anti-GD2 mAb	4 y	f	NBL, pd	Metastatic (jaw)	1st Relapse, PD	NA	A	short
#16	HR	anti-GD2 mAb	1 y	m	GNB	Metastatic (soft tissue, axilla)	Primary refractory, SD	A	D	normal
#17	HR	anti-GD2 mAb	8 y	m	NBL, pd	Metastatic (cns)	1st relapse (initial)	NA	B	long
#18	HR	anti-GD2 mAb	8 y	m	NBL, pd	Metastatic (lung)	1st relapse, PD	NA	E	n. d.

# indicates the sample IDs; anti-GD2 mAb: anti-GD2 monoclonal antibody; m/y: months/years; f: female; m: male; LR: low-risk; HR: high-risk; NBL: neuroblastoma; pd: poorly differentiated; dif: differentiating; GG: ganglioside; GNB: ganglioneuroblastoma; GN: ganglioneuroma; cns: central nervous system; NA: not amplified; A: amplified; n. d.: not determined; PD: progressive disease; SD: stable disease. Stage 4S is based on the International Neuroblastoma Staging System (INSS) [26]. “GG profile” and “ceramide” refer, respectively, to the predominant GG composition and the predominant length of the ceramide anchor identified in this study; ‘normal’ ceramide denotes the ceramide length typically found in healthy neurons. For more details, see Section 2.3. Samples 14 and 18 and samples 11 and 17 belong to the same patient and were retrieved at different time points.

**Table 2 ijms-26-08431-t002:** GG profiles of high-risk (HR) and low-risk (LR) neuroblastoma.

Profile	Prevalent GGs	Risk Status
A	b-series, particularly GT1b	All LR and some HR
B	GD2	HR
C	GM3 and or GD3	HR
D	GT1b and GD1a	HR
E	GD1a and GM2	HR

**Table 3 ijms-26-08431-t003:** Cell lines analyzed in this work.

Cell Line	Origin	Species	Subtype	Age	Sex	MYCN	ALK	GG Profile
NBL-S [28]	Adrenal tumor (primary)	Human	Adr [29]	36 m	male	No	No	E
LS [30]	Abdominal tumor	Human	Mes [31]	16 m	female	Yes	No	(E)
CHP-134 [32]	Left adrenal area	Human	Adr [27]	13 m	male	Yes	No	B
SH-SY5Y [33]	Bone marrow (subclon)	Human	Adr [27]	4 y	female	No	F1174L [34]	E
SK-N-AS [35]	Bone marrow (metastasis)	Human	Mes [27]	6 y	female	No	No	C
SK-N-BE(2)-C [36]	Bone marrow (subclon)	Human	Adr [27]	26 m	male	Yes	No	E
GI-ME-N [37]	Bone marrow (metastasis)	Human	Mes [27]	42 m	female	No	No	C
LA-N-5 [38]	Bone marrow (metastasis)	Human	Adr [27]	4 m	male	Yes	R1275Q [38]	(A)
ACN [39]	Bone marrow (metastasis)	Human	Mes [27]	36 m	male	No	No	C
IMR-32 [40]	Bone marrow (metastasis)	Human	Adr [27]	13 m	male	Yes	No	(E)
NXS2 [41]		Murine	-	-	-	unknown	unknown	E

In the category “Subtype”, Adr refers to adrenergic and Mes to mesenchymal cell state. “ALK” denotes the mutation status of the *ALK* gene; No indicates that no mutation was described in the cell line; “GG profile” refers to the GG composition identified in this study; m/y: months/years.

## Data Availability

Data are contained within the article or Appendix A. Survival and gene expression data of NBL RNA-seq and cell line datasets are accessible through the publicly available R2: Genomics Analysis and Visualization Platform (http://r2.amc.nl), accessed on 5 June 2025.

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
