# Peer review of "Ganglioside Profiling Uncovers Distinct Patterns in High-Risk Neuroblastoma"

_ijms, 2025, doi:10.3390/ijms26178431_

Round 1

Reviewer 1 Report

Comments and Suggestions for Authors

The paper "Ganglioside Profiling Uncovers Distinct Patterns in High-Risk Neuroblastoma" analyzed ganglioside expression profiles in neuroblastoma using mass spectrometry and thin-layer chromatography, revealing differences between high-risk and low-risk NBLs and proposing a risk stratification and treatment response prediction model based on the GG profile. The study design is rigorous and the data are comprehensive, but further improvement is needed in the following areas.
1. In the cell line analysis, some GG profiles (such as type D) did not have corresponding models, and this limitation needs to be clarified.
2. The relationship between CerS1/5/6 expression and sphingolipid anchor length, and whether GM3/GD3 enrichment reflects cancer stem cell characteristics, require additional immunohistochemistry or single-cell sequencing data.
3. Is GD2 downregulation common after anti-GD2 therapy? Paired samples before and after treatment need to be compared (currently only two cases are available). 4. Patient clinical characteristics should be supplemented with cross-analysis of MYCN amplification and GG profiles to clarify whether MYCN is involved. 5. The "ultra-high-risk (UHR) subgroup" mentioned is not defined in the results, and it is necessary to clarify whether the GG profile can assist in the identification of UHR.
6. Terminology consistency: The abbreviation "ganglioside" (GG or GSL) should be standardized throughout the text to avoid confusion.

Author Response

  1. In the cell line analysis, some GG profiles (such as type D) did not have corresponding models, and this limitation needs to be clarified.

Thank you for your comment. We have indeed analyzed only a small subset of the available NBL cell lines, and moreover GG profiles are not available from the literature for most of the cell lines. We have addressed this point starting from line 439: “About forty NBL cell lines have been described, but only a subset has been characterized for their GG profiles. Further GG profiling is needed to identify cell lines that match all GG profiles observed in patients samples. Such cell lines would enhance the available in vitro models, particularly for studying aggressive NBL subtypes [75].

  1. The relationship between CerS1/5/6 expression and sphingolipid anchor length, and whether GM3/GD3 enrichment reflects cancer stem cell characteristics, require additional immunohistochemistry or single-cell sequencing data.

We agree that the next steps would be to confirm the relevance of our finding with other methods. We have added it in the discussion (lane 357 and lane 338)

“A correlation between CerS1/5/6 expression and ceramide length in NBL, as well as the association of CerS expression with prognosis, should be analyzed in larger cohorts.”

Further studies are also needed to determine whether GM3/GD3 enrichment is indicative of cancer stem cell–like characteristics in NBL.”

  1. Is GD2 downregulation common after anti-GD2 therapy? Paired samples before and after treatment need to be compared (currently only two cases are available).

We have indeed only two paired samples before and GD2 therapy (14/18 and 11/17). Both samples show a reduction in GD2 after anti-GD2 therapy. Loss of GD2 as mechanism of resistance can however account for only a portion of anti-GD2 antibody resistance. We have improved this topic in the discussion (lane 397): “While a low amount of GD2 has been discussed as a reason for resistance to anti-GD2 mAbs [59], only 10–12% of NBL patients show complete or partial loss of GD2 expression in bone marrow–derived NBL cells [60], a percentage far lower than the overall rate of non-response to antibody therapy. Therefore, constitutively low or absent GD2 expression can account for only a portion of anti-GD2 directed therapies resistance. Indeed, a recent study using anti-GD2 CAR-T cells found no loss of GD2 expression on relapsed tumor cells after treatment failure [14].”

  1. Patient clinical characteristics should be supplemented with cross-analysis of MYCN amplification and GG profiles to clarify whether MYCN is involved.

We did not find any correlation between MYCN amplification and a particular GG profile or ceramide length (Chi-square test). We have added the result in the supplemental Table 3 and 4. However, of course the number of samples is too low to allow solid correlations

  1. The "ultra-high-risk (UHR) subgroup" mentioned is not defined in the results, and it is necessary to clarify whether the GG profile can assist in the identification of UHR.

In our small cohort, we had two patients with profile C with an extraordinarily aggressive clinical course (lane 245), we discuss now if this profile may identify  ultra-high-risk patients (lane 335)

"Both patients had an extraordinarily aggressive clinical course and died relatively early after diagnosis due to uncontrolled progressive disease"

This profile was found in patients with an exceptionally severe clinical course who died of uncontrolled progressive disease early after primary diagnosis in our small cohort. Therefore, the biosynthetic stop beyond GM3 and GD3 with consequent minimal GD2 expression may characterize a highly aggressive, immature NBL subtype, constituting an ultra-high-risk cohort, resistant to GD2 targeted immunotherapeutic approaches. This finding should be necessarily validated in a larger cohort. 

  1. Terminology consistency: The abbreviation "ganglioside" (GG or GSL) should be standardized throughout the text to avoid confusion.

GG refers only to gangliosides,. Gangliosides belong to the Glycosphingolipids (GSL) family. We have corrected a confounding definition at position 100

Reviewer 2 Report

Comments and Suggestions for Authors

Ganglioside Profiling Uncovers Distinct Patterns in High-Risk Neuroblastoma by Paret et al. In this Ms, the author analyzed the GG profiles of 18 tumor samples from patients and 11 NBL cell lines using thin-layer chromatography and mass spectrometry. The expression of 0-, a-, and b-series GG was detected, and it was correlated with clinical risk, outcomes, and gene expression data. The results showed that the composition of GG and the expression of related enzymes could serve as potential biomarkers for NBL risk stratification and treatment response. The study findings are helpful for developing targeted treatment strategies and clinical practice for NBL. The manuscript is well-written and I have no major objections to the research results. I think the manuscript can be accepted and published after a minor revision. To ensure the publication readiness of the manuscript, the following issue needs to be addressed:

Comments:

  1. In the introduction, briefly summarize the biosynthesis of GG, especially the research involving B4GALNT1 in NBL.
  2. Sections 4.3 and 4.4 of the Materials and Methods should provide more detailed descriptions.

Author Response

  1. In the introduction, briefly summarize the biosynthesis of GG, especially the research involving B4GALNT1 in NBL.

Thank you for this suggestion. We have briefly summarize the biosynthesis of GG at lane  80 and commented on B4GALNT1 in NBL at position 92.

  1. Sections 4.3 and 4.4 of the Materials and Methods should provide more detailed descriptions.

We have added more details for both methods in the respective sections

Reviewer 3 Report

Comments and Suggestions for Authors

This study investigates ganglioside (GG) profiles in neuroblastoma (NBL) to identify biomarkers for risk stratification and therapy response. Using mass spectrometry and TLC, the authors analyzed 18 patient-derived tumors and 11 cell lines, revealing distinct GG patterns between low-risk (LR) and high-risk (HR) NBL. Key findings include: (1) LR tumors exhibit enriched b-series GG (e.g., GT1b), while HR tumors show heterogeneous profiles (A–E); (2) Profile C (GM3/GD3 dominance due to B4GALNT1 downregulation) correlates with poor prognosis; (3) Ceramide anchor length of GD2 differs in HR vs. LR tumors; and (4) Anti-GD2 therapy resistance associates with reduced GD2 or shorter ceramide anchors. The study’s strengths are its clinical relevance, multi-omics integration (GG profiling, gene expression, survival analysis), and potential for refining NBL subclassification.  

Minor Points
1. The biosynthesis figure (Fig. 1) is critical but not included. Ensure symbols (e.g., "a-", "b-" series) are consistently defined in the legend.  
2. Samples #14 and #18 are from the same patient but listed as separate entries. Clarify if these are distinct lesions/time points.  
3. Figure 3 Panel B: "Y-axis" label missing (ratio of GM3+GD3 / other GG). Clarify why MYCN-amplified samples (red) cluster separately.  
4. Figure 4B: Forest plot lacks cohort details (e.g., treatment status). Clarify if B4GALNT1 low expression was prognostic across all HR subtypes.  
5. The link between GG profiles and retinoid sensitivity is speculative. Cite prior evidence (e.g., ATRA-induced GG changes in vitro).  
6. "Small number of samples" is acknowledged but downplayed. Emphasize this as a key limitation. 
7. Specify internal standards used for each GG. The MRM transitions (Suppl. Table 1) should be referenced in the main text.  
8. For meta-analysis, detail how B4GALNT1 "low expression" was defined (quartiles? continuous?). The *p*-value for heterogeneity (Cochran’s Q) is reported but not the effect size (τ²).  
9. While B4GALNT1 is proposed as a prognostic marker, its predictive value for anti-GD2 response remains untested. The "ultra-high-risk" subgroup (UHR) is mentioned but not defined by GG profiles.  
10. The study correlates GG profiles with clinical outcomes but does not mechanistically test how specific GG (e.g., short-ceramide GD2) drive aggressiveness or therapy resistance.  

Author Response

  1. 1. The biosynthesis figure (Fig. 1) is critical but not included. Ensure symbols (e.g., "a-", "b-" series) are consistently defined in the legend.  

We have improved the description of the biosynthesis and the legend of Figure 1.

2. Samples #14 and #18 are from the same patient but listed as separate entries. Clarify if these are distinct lesions/time points.  

The samples are from different time points, we have clarified this in the legend

  1. Figure 3 Panel B: "Y-axis" label missing (ratio of GM3+GD3 / other GG). Clarify why MYCN-amplified samples (red) cluster separately.  

We have included the label in the legend.

MYCN samples seems to have a bigger ratio in the alive group. However, this was not significant (data not shown). MYCN status also did not associate to a particular GG profile (Supplemental Table 3 and 4), but obviously the number of samples is too low and to heterogeneous to allow a solid statistic (we have addressed this in the discussion, lane 456)

  1. Figure 4B: Forest plot lacks cohort details (e.g., treatment status). Clarify if B4GALNT1 low expression was prognostic across all HR subtypes.  

We updated the Figure 4B by adding the sample and event number. Also, we restricted the Westermann dataset to diagnostic only (disease class: initial) samples to avoid bias from post-treatment/rlapse sampling. This improved cross-cohort comparability and reduced heterogeneity (Q from 1.78 to 0.13; τ² upper bound from 0.78 to 0.049), while strengthening the pooled association (HR 0.73 → 0.67). This supports a baseline-prognostic role of B4GALNT1. Bell and SEQC did not report treatment status at sampling.

We added a high-risk-only meta-analysis (Westermann & SEQC). Bell study did not reported the risk status and was excluded. The pooled HR was 0.84 (0.62–1.14; k=2), directionally consistent but not statistically significant due to reduced sample size and events. The result was added to the Supplemental Table 5

  1. The link between GG profiles and retinoid sensitivity is speculative. Cite prior evidence (e.g., ATRA-induced GG changes in vitro).  

We thanks the reviewer for this observation. Indeed, the correlation between GG expression, differentiation and interaction with ATRA is complex. Although it has been shown that GG expression can drive NBL differentiation and that ATRA induces the synthesis of complex GG, the mechanisms are not fully clarified. We have included more literature in the discussion, also literature that shows that ATRA can induce differentiation even in the absence of GG, leading to the conclusion that mechanistic studies are needed to assess the relevance of GG profiles in resistance to retinoid therapy

6. "Small number of samples" is acknowledged but downplayed. Emphasize this as a key limitation. 

We have added that the small number is a key limitation, and added the heterogeneity as limitation (lane 456)

7. Specify internal standards used for each GG. The MRM transitions (Suppl. Table 1) should be referenced in the main text.  

We have referenced the MRM transitions and specified the internal standards in material and methods

  1. For meta-analysis, detail how B4GALNT1 "low expression" was defined (quartiles? continuous?). The *p*-value for heterogeneity (Cochran’s Q) is reported but not the effect size (τ²).  

The meta-analysis pools continuous B4GALNT1 coefficients. We clarified this aspect in the Methods Section 4.8 and in the results. We now report complete heterogeneity statistics: τ² = 0.00 (95% CI 0.00–0.78), I² = 0.0% (0.0–89.6), Q(2) = 1.78, p = 0.41 in the results.

  1. While B4GALNT1 is proposed as a prognostic marker, its predictive value for anti-GD2 response remains untested. The "ultra-high-risk" subgroup (UHR) is mentioned but not defined by GG profiles.  

Unfortunately, we did not have access to datasets containing survival data in which patients were stratified according to anti-GD2 therapy status. As a result, we were unable to directly assess the predictive value of B4GALNT1 expression for treatment response. We acknowledge this as a limitation of the study and have now explicitly noted it in the revised manuscript (lane 255: “Unfortunately, we did not have access to datasets containing survival data in which pa-tients were stratified according to anti-GD2 therapy status. As a result, we were unable to directly assess the predictive value of B4GALNT1 expression for treatment response”).

In our small cohort, we had two patients with profile C with an extraordinarily aggressive clinical course (lane 245), we discuss now if this profile may identify  ultra-high-risk patients (lane 332)

  1. The study correlates GG profiles with clinical outcomes but does not mechanistically test how specific GG (e.g., short-ceramide GD2) drive aggressiveness or therapy resistance.  

We agree that mechanistically test should be performed in the future, and we have added this to the discussion (lane 425: Moreover, mechanistically test in the future would be necessary to verify how specific GG, including GD2 with different ceramide lengths, drive aggressiveness or therapy resistance in NBL )